# The Role of Brownmillerite in Preparation of High-Belite Sulfoaluminate Cement Clinker

Xuemei Chen [1,2], Jun Li [1,*], Zhongyuan Lu [1,*], Serina Ng [3], Yunhui Niu [1], Jun Jiang [1], Yigang Xu [2], Zhenyu Lai [1] and Huan Liu [3]

1 State Key Laboratory of Environment-Friendly Energy Materials, School of Material Science and Engineering, Southwest University of Science and Technology, Mianyang 621010, China; chunruxue@126.com (X.C.); niuyunhui@swust.edu.cn (Y.N.); jiangjun@swust.edu.cn (J.J.); laizhenyu@swust.edu.cn (Z.L.)
2 Jiahua Special Cement Co., Ltd., No. 2 Ma'anshan Jiufeng Rd, Shizhong District, Leshan 614003, China; zhuasc@163.com
3 Shijiazhuang Chang'An Yucai Building Materials Co., Ltd., No. 11 Fucheng Rd., Shijiazhuang 051430, China; serina.ng@sjzcayc.com (S.N.); youyuanzhuren@163.com (H.L.)
* Correspondence: lijun@swust.edu.cn (J.L.); luy@swust.edu.cn (Z.L.); Tel./Fax: +86-816-2419212 (J.L.)

**Abstract:** High-belite sulfoaluminate cement (HBSC) clinker containing brownmillerite was prepared using the industrial raw materials limestone, aluminum tailings, aluminum ore waste rock, and anhydrite. The effect of brownmillerite on clinker sintering and clinker minerals and the mechanical performance of HBSC was investigated using thermal analysis, petrographic analysis, and quantitative X-ray diffraction (QXRD). Results indicated that brownmillerite promoted the formation of clinker minerals and stabilized calcium sulfoaluminate ($C_4A_3\$$) through the substitution of $Fe^{3+}$ for $Al^{3+}$ in $C_4A_3\$$, which increased the actual $C_4A_3\$$ content and decreased the brownmillerite content compared to that of the designed theoretical mineral composition. However, the early compressive strength of HBSC pastes decreased with the increase in brownmillerite content due to the decrease in the total amount of early-strength clinker minerals. Brownmillerite also influenced belite ($C_2S$) structures and increased the $\gamma$-$C_2S$ content with poor hydration activity, thus inhibiting the strength development of HBSC pastes. The proper amount of brownmillerite in HBSC clinker can ensure the early strength and strength development of HBSC pastes.

**Keywords:** brownmillerite; high-belite sulfoaluminate cement clinker; mineral composition; compressive strength





## 1. Introduction

The production of Portland cement clinker generates huge carbon emissions due to the combustion of fossil fuel and decomposition of limestone. It is not easy to further reduce carbon emissions through energy conservation and emission reduction technologies for traditional Portland cement production. Developing new cement clinkers with a lower sintering temperature and CaO content seems to be one of the most important directions for carbon emission reduction in the cement industry [1–4].

The main minerals in Portland cement are alite ($C_3S$), belite ($C_2S$), tricalcium aluminate ($C_3A$), and ferrite solid solution (brownmillerite). Among these, the formation of $C_2S$ minerals requires lower energy and limestone consumption. In particular, the hydration rate and heat release of $C_2S$ are obviously lower than other clinker minerals. Therefore, clinker with $C_2S$ as the dominant mineral has been considered an ideal green low-carbon cementitious material due to its low content of CaO, ultralow hydration heat, high later mechanical performance, and high corrosion resistance [5]. However, the low early hydration rate of $C_2S$ leads to the lower early mechanical performance of cement, which significantly decreases the construction speed. Researchers have tried to activate or stabilize higher hydration activity $C_2S$ through sintering parameter adjustment or mineral/ion doping, but

few results have been satisfactory. In fact, a third series of cement, named sulfoaluminate cement, was invented in the 1970s. Anhydrous calcium sulfoaluminate ($C_4A_3\$$) is the main mineral in sulfoaluminate cement clinker. Sulfoaluminate cement clinker also has low calcination temperature and CaO content, which effectively reduces the use of fossil fuel and limestone [6]. At present, sulfoaluminate cement has been successfully used in emergency repair and construction projects, anti-seepage and plugging applications, construction in winter, GRC products, and other construction projects [7,8], because of its fast setting, high mechanical performance, low alkalinity, microexpansion and corrosion resistance. However, a high-quality bauxite requirement, high expansibility and unstable strength growth have limited the sulfoaluminate cement production and application. Based on the characteristics of $C_2S$ and $C_4A_3\$$ minerals, the combination of $C_2S$ and $C_4A_3\$$ minerals in a high-belite sulfoaluminate cement (HBSC) system has been proposed to overcome the respective shortcomings of belite cement and sulfoaluminate cement [9–14]. It is well known that an appropriate amount of brownmillerite can promote the sintering reaction and reduce the calcination temperature. In industrial-scale production, the aluminum material quality for HBSC is lower than that for sulfoaluminate cement, and higher $Fe_2O_3$ usually exists in aluminum materials. Therefore, in this study, a large number of brownmillerite phases will be introduced in the firing process of HBSC. At the present time, the effects of tetracalcium ferrite ($C_4AF$) and brownmillerite on the sintering and properties of Portland cement clinker, high belite cement clinker and sulfoaluminate cement clinker have already been researched [15–19]. Some researchers pointed out that brownmillerite mainly influenced the later strength, while others believed that the role of brownmillerite in various types of cement should be correctly evaluated and understood. A poor cementitious property of brownmillerite was observed in high-alkalinity Portland cement paste, but its strength contribution was greater in low-alkalinity cement paste [20]. However, the role of brownmillerite in the preparation of HBSC clinker has not been made clear until now.

This study aims to prepare a kind of green and low-carbon HBSC. Currently, the related research on high-belite sulfoaluminate cement is still in the laboratory stage, and it cannot be fully applied in production. The main reason is that the differing grades of industrial raw materials, especially changes in iron content, greatly influence its production and performance. In this study, low-grade aluminum material with high iron content was selected to investigate its influence on HBSC because it can improve the material selection and adaptation range for HBSC production. It can also promote this technology application to fill the gap between scientific research and application.

In this study, HBSC clinkers containing different contents of brownmillerite were prepared. The role of brownmillerite in preparing HBSC clinker was investigated to provide references for the industrial production of HBSC clinker with high brownmillerite content.

## 2. Experimental Section

### 2.1. Materials

Industrial raw materials, including limestone (LS), aluminum tailings (AT) which were the residual material of aluminum ore for sintering ceramics, waste rocks of aluminum mine (AM) that were the waste rock of bauxite, and anhydrite (AH), were all provided by Qianghua Special Cement Engineering Co., Ltd., Sichuan Province, China. The main chemical compositions and trace elements of raw materials are shown in Tables 1 and 2, respectively.

It can be seen from Table 1 that the content of $SiO_2$ in AT is higher than that of bauxite ($SiO_2$ < 15.0 wt%) used for sulfoaluminate cement, and the content of $Al_2O_3$ is much lower than that of bauxite ($Al_2O_3$ > 55.0 wt%) used for sulfoaluminate cement. Similarly, the content of $SiO_2$ in AM is higher and the content of $Al_2O_3$ is lower. At the same time, its content of $Fe_2O_3$ is higher ($Fe_2O_3$ > 20%) than that of conventional bauxite ($Fe_2O_3$ < 10%) used for sulfoaluminate. Both materials are inferior and unsuitable in Portland cement and sulfoaluminate cement. However, they are just suitable for preparing HBSC. At the same time, they contain some trace elements conducive to sintering, such as Cuo, ZnO, SrO, Bao, etc., and harmful elements (Cl, $Cr_2O_3$, MnO etc.) are very low, as shown in Table 2.



**Table 1.** Main chemical compositions of raw materials.

| Materials | Loss | SiO$_2$ | Fe$_2$O$_3$ | Al$_2$O$_3$ | CaO | MgO | TiO$_2$ | SO$_3$ | $\sum$ |
|-----------|------|---------|-------------|-------------|------|------|---------|--------|--------|
| LS | 43.37 | 1.03 | 0.16 | 0.26 | 54.13 | 0.44 | - | - | 99.39 |
| AT | 14.06 | 37.20 | 1.88 | 43.15 | 0.57 | 0.41 | - | 0.05 | 97.32 |
| AM | 9.70 | 39.18 | 24.46 | 19.74 | 1.69 | 0.60 | 3.87 | - | 99.24 |
| AH | 8.47 | 4.05 | 1.02 | 0.41 | 33.85 | 5.12 | - | 46.60 | 99.52 |

**Table 2.** Trace elements of raw materials.

| Materials | Cl | Cr$_2$O$_3$ | MnO | NiO | CuO | ZnO | SrO | ZrO$_2$ | BaO |
|-----------|------|-------------|-------|-------|-------|-------|-------|---------|-------|
| LS | 0.004 | 0.010 | 0.082 | 0.015 | 0.028 | 0.022 | 0.019 | 0.143 | 0.065 |
| AT | 0.002 | 0.014 | 0.068 | 0.020 | 0.007 | 0.020 | 0.028 | 0.065 | 0.062 |
| AM | 0.010 | 0.009 | 0.059 | 0.010 | 0.021 | 0.015 | 0.036 | 0.237 | 0.057 |
| AH | 0.006 | 0.003 | 0.000 | 0.004 | 0.005 | 0.003 | 0.408 | 0.000 | 0.020 |

*2.2. Preparation*

2.2.1. Mineral Design of Clinker

Compared to sulfoaluminate cement clinker, whose mineral content is mainly anhydrous calcium sulfoaluminate (above 55 wt%) and a small content of brownmillerite (below 5 wt%), the mineral composition of high-belite-sulfoaluminate cement (HBSC) clinker containing brownmillerite, as used in this study, is different. Its mineral content is mainly C$_2$S (above 45 wt%) and the content of brownmillerite is more than 10 wt%. So, the traditional proportioning rate range and calculation formula cannot be applied, and thus, the mineral composition of HBSC clinker was directly designed, taking into account the increased content of brownmillerite. The chemical composition was calculated by optimizing the Bogue formula [21]. Detailed principles are assumed as follows: (1) all Fe$_2$O$_3$ forms brownmillerite phase 4CaO·Al$_2$O$_3$·Fe$_2$O$_3$ (C$_4$AF); (2) all Al$_2$O$_3$ forms 4CaO·3Al$_2$O$_3$·SO$_3$ (C$_4$A$_3$\$) except for the brownmillerite phase; (3) remaining CaSO$_4$ exists in the form of free CaSO$_4$ (*f*-CS); (4) all SiO$_2$ forms 2CaO·SiO$_2$ (C$_2$S, belite); (5) residual CaO exists in the form of free CaO (*f*-CaO). Based on the above assumptions, the mineral composition of clinker was calculated according to Equations (1)–(4):

$$w(\text{C}_4\text{AF}) = 3.04 * w(\text{Fe}_2\text{O}_3) \tag{1}$$

$$w(\text{C}_4\text{A}_3\$) = 1.995 * [w(\text{Al}_2\text{O}_3) - 0.641 * w(\text{Fe}_2\text{O}_3)] \tag{2}$$

$$w(f\text{-CS}) = 1.70 * [w(\text{SO}_3) - 0.26 * w(\text{Al}_2\text{O}_3) + 0.17 * w(\text{Fe}_2\text{O}_3)] \tag{3}$$

$$w(\text{C}_2\text{S}) = 2.87 * w(\text{SiO}_2) \tag{4}$$

In order to study the influence of brownmillerite upon HBSC, the content of C$_2$S and free gypsum in HBSC clinker shall be controlled to be consistent according to the characteristics of the raw materials, and the brownmillerite shall be adjusted to increase by 5.0%. According to the calculation of batching formula, the increase in brownmillerite content will be accompanied by a decrease in anhydrous calcium sulfoaluminate. In order to maintain physical properties, the content of anhydrous calcium sulfoaluminate should not be too low (>10.0 wt%). The mix design of the HBSC raw materials and the theoretical mineral composition of the designed HBSC clinker are shown in Table 3.

**Table 3.** Mix design of raw materials and theoretical mineral composition of HBSC clinker.

| Sample | Mix Design of Raw Materials (wt%) | | | | Minerals Composition (wt%) | | | |
|---|---|---|---|---|---|---|---|---|
| | LS | AM | AH | AT | $C_4A_3\$$ | $C_2S$ | $C_4AF$ | $f$-CS |
| GF-1 | 60.6 | 12.4 | 7.0 | 20.0 | 28.79 | 51.50 | 15.18 | 1.70 |
| GF-2 | 61.2 | 17.8 | 6.0 | 15.0 | 24.26 | 51.00 | 20.24 | 1.56 |
| GF-3 | 62.0 | 23.0 | 5.0 | 10.0 | 19.69 | 50.29 | 25.14 | 1.44 |
| GF-4 | 62.8 | 28.2 | 4.0 | 5.0 | 15.09 | 49.57 | 30.07 | 1.31 |
| GF-5 | 63.1 | 33.9 | 3.0 | 0.0 | 10.57 | 49.39 | 35.45 | 1.16 |

### 2.2.2. Clinker Preparation

Raw materials, including LS, AT, AM, and AH, were dried and separately ground by a $\varphi 500 \times 500$ mm test ball, according to the reference standard GB/T 26567-2011 test method for the grindability of cement raw materials-bond method. The 80 μm sieve residue was less than 10% for all raw material powders, according to the reference standard GB/T 26566-2011 test method for the burnability of cement raw meal. Powders of raw materials were weighted according to the proportioning design in Table 1 and evenly mixed. Then, 10 wt% water of solids was added and mixed with powders, and the wet powders were then pressed into a $\varphi 120$ mm $\times$ 10 mm waveform cake. Cakes dried in an oven at $105 \pm 5\ °C$ to a constant weight were then placed on a platinum sheet and sintered in a high-temperature electric furnace at the set temperature for 30 min. After sintering, the cakes were taken out quickly and cooled to room temperature to obtain the cement clinker.

### 2.2.3. Cement Preparation

The prepared clinkers were mixed with 8 wt% anhydrite, then ground into powders with a specific surface area of $350 \pm 10$ m$^2$/kg by a vibration mill to obtain HBSC. HBSC pastes were prepared by using a water-to-binder ratio of 0.40. Pastes were molded in $20 \times 20 \times 20$ mm steel molds, then cured in a curing box at 20 °C and 95% RH for designated periods.

### 2.3. Characterization

X-ray diffraction (XRD) patterns were collected on a Germany Bruker D8 ADVANCE diffractometer with Cu Ka radiation ($\lambda = 1.5406$ Å) generated at room temperature at 60 kV and 80 mA. Powders were step-scanned from 5° to 70° with a step size and time per step of 0.02° and 0.5 s, respectively. Identification of clinker minerals was implemented by analysis of EVA software. Rietveld refinement quantitative phase analysis was performed by using TOPAS software. The refined overall parameters included emission profile, background, instrument factors, and zero error. The Rwp value of the profile refinement was used to evaluate the quality of the fits in the Rietveld refinement process. Generally, the result was considered reliable if the Rwp value was less than 10% [22]. The ICSD codes of the main minerals in BBSC clinker in this study are shown in Table 4 [22,23].

**Table 4.** ICSD of the main minerals in BBSC clinker and its hydration products in this study.

| Mineralogical Phase | ICSD |
|---|---|
| $\alpha$-$C_2S$ | 81097 |
| $\beta$-$C_2S$ | 81096 |
| $\gamma$-$C_2S$ | 81095 |
| $C_4A_3\$$ | 9560 |
| $C_4AF$ | 9197 |
| Perovskite | 62149 |
| $C_2AS$ | 87144 |
| Anhydrite | 16382 |
| $C_2F$ | 27808 |
| CF | 16695 |

Thermal analysis of the sample was conducted using thermogravimetry and differential scanning calorimetry (TG-DSC, STA449F3Jupiter) produced by Netzsch Group of Germany, at a temperature ranging from 30 °C to 1400 °C in a nitrogen atmosphere with the rate of temperature increase standing at 10 °C/min.

In order to better observe the internal and representative mineral structure of the clinker, it was necessary to break the clinker test cake into samples with a particle size of less than 5mm. After the clinkers were crushed, we put the representative sample into the molten but unburned sulfur for soaking, and then poured them into the specific mold to complete the inlaying of clinker samples and prepare the lithofacies light sheet. Then, the lithofacies light sheet was ground by the grinding and polishing machine until the surface was as bright as a mirror without scratches. In addition, 1% ammonium chloride solution was used to erode the clinker, and mineral morphology was observed after erosion [24]. The mineral morphology of clinkers was analyzed using the research-grade upright digital material microscope (Axio Scope.A 1) produced by CARL ZEISS of Germany.

### 3. Results and Discussion

#### 3.1. Thermal Evolution of HBSC Raw Materials

TG/DTA curves of HBSC raw materials (set out in Table 1) are shown in Figure 1; the corresponding data of characteristic peaks in the DTA and TG curves are listed in Tables 5 and 6, respectively. There is an endothermic peak (Figure 1a) and slight weight loss (Figure 1b) between 100 °C and 130 °C, which was attributed to physical water evaporation. The second endothermic peak (Figure 1a) associated with an obvious weight loss (Figure 1b) located at 400–600 °C was attributed to the decomposition of boehmite ($\gamma$-AlOOH) in aluminous materials. The highest endothermic peak (Figure 1a), associated with the maximum weight loss (Figure 1b) located at 700–900 °C, is observed, which corresponded to the decomposition of LS. From 950 °C to 1280 °C, many of the endothermic peaks in Figure 1a appeared without weight losses in Figure 1b. From 1280 °C to 1330 °C, an exothermic peak appeared without weight losses, which was the clinker mineral formation stage accompanied by complex solid-phase reaction processes, including the formation of anhydrous $C_4A_3\$$, $C_2S$, and brownmillerite $C_4AF$. It was noted that the endothermic temperature would be slightly decreased with the increase in designed brownmillerite content, as shown in Table 5. It can be seen from the DTA curve that the reaction peak at temperatures from 1280 °C to 1330 °C becomes evident with the increase in brownmillerite content. When the temperature increased above 1330 °C, another endothermic peak associated with weight loss should indicate the decomposition of some clinker minerals, especially the sulfoaluminate phases. As shown in Table 6, weight loss in this stage except for GF-1 decreased with the increase in designed brownmillerite content, and the weight loss was lower than that of the sample with brownmillerite content of less than 20% (GF-1), which preliminarily showed that the brownmillerite content could stabilize the anhydrous calcium sulfoaluminate and reduce its decomposition. According to the above analysis, the sintering temperature of HBSC clinker was mainly between 1200 °C and 1400 °C.

**Table 5.** Key temperature point of exothermic or endothermic peaks in DTA curves (°C).

| Sample | Peak 1 | Peak 2 | Peak 3 | Peak 4 |
| --- | --- | --- | --- | --- |
| GF-1 | 125.4 | 516.5 | 863.1 | 1241.9 |
| GF-2 | 121.9 | 512.2 | 864.5 | 1239.3 |
| GF-3 | 121.8 | 512.3 | 864.2 | 1238.1 |
| GF-4 | 120.7 | 510.5 | 860.4 | 1237.3 |
| GF-5 | 118.0 | 507.5 | 863.6 | 1235.8 |

**Table 6.** Weight loss of HBSC raw materials in TG curves (%).

| Sample | Stage I | Stage II | Stage III |
|--------|---------|----------|-----------|
| GF-1 | 4.65 | 25.63 | 0.89 |
| GF-2 | 4.51 | 26.63 | 1.08 |
| GF-3 | 3.70 | 26.84 | 0.89 |
| GF-4 | 3.26 | 27.30 | 0.65 |
| GF-5 | 2.87 | 28.48 | 0.58 |

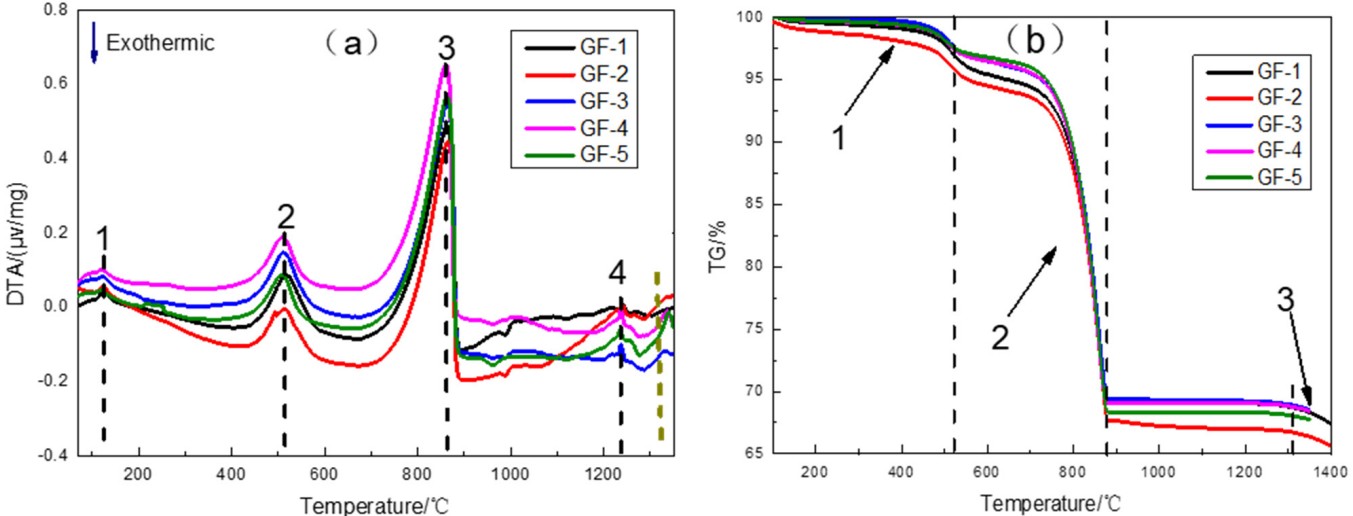

**Figure 1.** (**a**) DTA (1-Peak 1, 2-Peak2, 3-Peak3, 4-Peak4) and (**b**) TG (1-Stage I, 2-Stage II, 3-Stage III) curves of HBSC raw materials.

### 3.2. Sintering of HBSC Clinker Containing Brownmillerite

3.2.1. Burnability

The observed physical appearance of HBSC clinker is summarized in Table 7. According to the TG-DTA results, sintering of HBSC clinker containing brownmillerite at 1300 °C, 1320 °C, 1330 °C and 1350 °C for 30 min was proposed, which was also used to evaluate the burnability of HBSC raw materials. The sintering temperature of clinker with proper appearance quality decreased with the increase in designed brownmillerite content. Normal and dense clinker was obtained at 1300 °C when the designed brownmillerite content in HBSC clinker exceeded 20%. Under these conditions, the clinker was hard and fully calcined. Higher sintering temperatures above 1300 °C led to over-burnt or molten HBSC clinker with higher designed brownmillerite content, causing the clinker to stick to the platinum sheet. According to the appearance of HBSC clinkers sintering at different temperatures, the shrinkage of clinkers GF-1, GF-2 and GF-3 was normal and dense at 1330 °C, while clinkers GF-4 and GF-5 were slightly over-calcined, leading to slight sticking, which could be removed from the platinum sheet. Most clinkers were seriously over-calcined and molten and adhered to the platinum sheet and proved hard to remove when the sintering temperature exceeded 1330 °C. In additiona sintering temperature below 1330 °C led to the under-burning of HBSC clinker. It can be seen from Figure 2 that the color of the clinker gradually changed from dark brown to light black with the increase in designed brownmillerite content, and the clinker became denser and glossier in this case. HBSC clinkers also became easier to melt with the increase in designed brownmillerite content.

**Table 7.** Physical appearance observations of HBSC clinker.

| Sample | 1350 °C | 1330 °C | 1320 °C | 1300 °C |
|---|---|---|---|---|
| GF-1 | Reddish brown, highly dense | Reddish brown, dense | Reddish brown, loose | Reddish brown, loose |
| GF-2 | Brown, sticky | Brown, dense | Brown, dense | Brown, loose |
| GF-3 | Gray, highly sticky | Gray, dense | Gray, dense | Gray, dense |
| GF-4 | Dark gray, highly sticky | Dark gray, slightly sticky | Dark gray, dense | Dark gray, dense |
| GF-5 | Light black, highly sticky | Light black, sticky | Light black, dense | Light black, very dense |

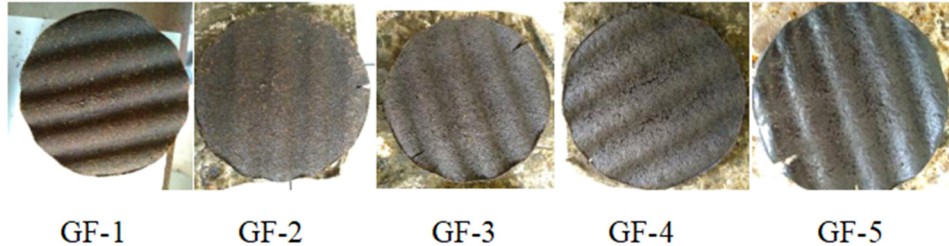

**Figure 2.** Photos of HBSC clinker appearance sintering at 1330 °C.

$f$-CaO of HBSC clinker containing brownmillerite sintering at different temperatures was detected. The results are shown in Table 8. $f$-CaO of the HBSC clinker sintering at 1300 °C was lower than 1.0% due to the formation of a large number of clinker minerals. The content of $f$-CaO in HBSC clinker decreased gradually with the increase in sintering temperature. Less than 0.30% of $f$-CaO was detected when the sintering temperature was elevated to 1330 °C, which indicated that the clinker had been completely sintered. As observed in Figure 1b, sulfoaluminate phases would decompose above 1300 °C, which increased the $f$-CaO in the HBSC clinker. However, $f$-CaO in HBSC clinker sintering at 1350 °C, except for GF-1 and GF-2, was further decreased compared with HBSC clinker sintering at 1330 °C. This finding demonstrated that brownmillerite had a particular stabilizing effect on anhydrous calcium sulfoaluminate when the designed brownmillerite content in HBSC clinker was greater than 20%. Based on the above, we concluded that HBSC raw materials designed in this study had better burnability, especially the samples sintering at 1330 °C.

**Table 8.** Content of $f$-CaO in HBSC clinker sintering at different temperatures (%).

| Sample | 1300 °C | 1320 °C | 1330 °C | 1350 °C |
|---|---|---|---|---|
| GF-1 | 0.92 | 0.41 | 0.29 | 0.32 |
| GF-2 | 0.67 | 0.57 | 0.00 | 0.29 |
| GF-3 | 0.93 | 0.53 | 0.23 | 0.15 |
| GF-4 | 0.93 | 0.73 | 0.26 | 0.22 |
| GF-5 | 0.64 | 0.70 | 0.26 | 0.13 |

### 3.2.2. Mineral Morphology

Petrographic images of HBSC clinker sintering at 1330 °C for 30 min are shown in Figure 3. These show mainly small, tree leaf-like and round, grain-like mineral particles in HBSC clinkers. Most mineral particles were blue, though some were brown. Some minerals had parallel crystal lines under the erosion of ammonium chloride solution. Tree leaf-like minerals with smooth edges and sizes in the length of 10–60 μm were identified as $C_2S$ based on morphological analysis [25–27]. Relatively uniform round particles with sizes of 2–5 μm could also be identified as $C_2S$. The content of $C_2S$ increased with the increase in designed brownmillerite content. White round minerals that appeared in the clinker with sizes of 5–10 μm could be identified as brownmillerite, which increased with the increase in designed brownmillerite content (Figure 3d,e). Gray minerals with regular tetragonal and hexagonal shapes or fragmented particles could be identified as anhydrous $C_4A_3\$$.

Petrographic analysis showed that the high content of brownmillerite had an obvious impact on the morphology of HBSC clinker minerals. $C_2S$ was found mainly as tree leaf-like and round particles, and the leaf-like $C_2S$ increased with the increase in designed brownmillerite content. When the designed brownmillerite content exceeded 25%, there was an apparent white round brownmillerite mineral in the clinker, which also increased with the increase in designed brownmillerite content, and the mineral nest appeared in this case.

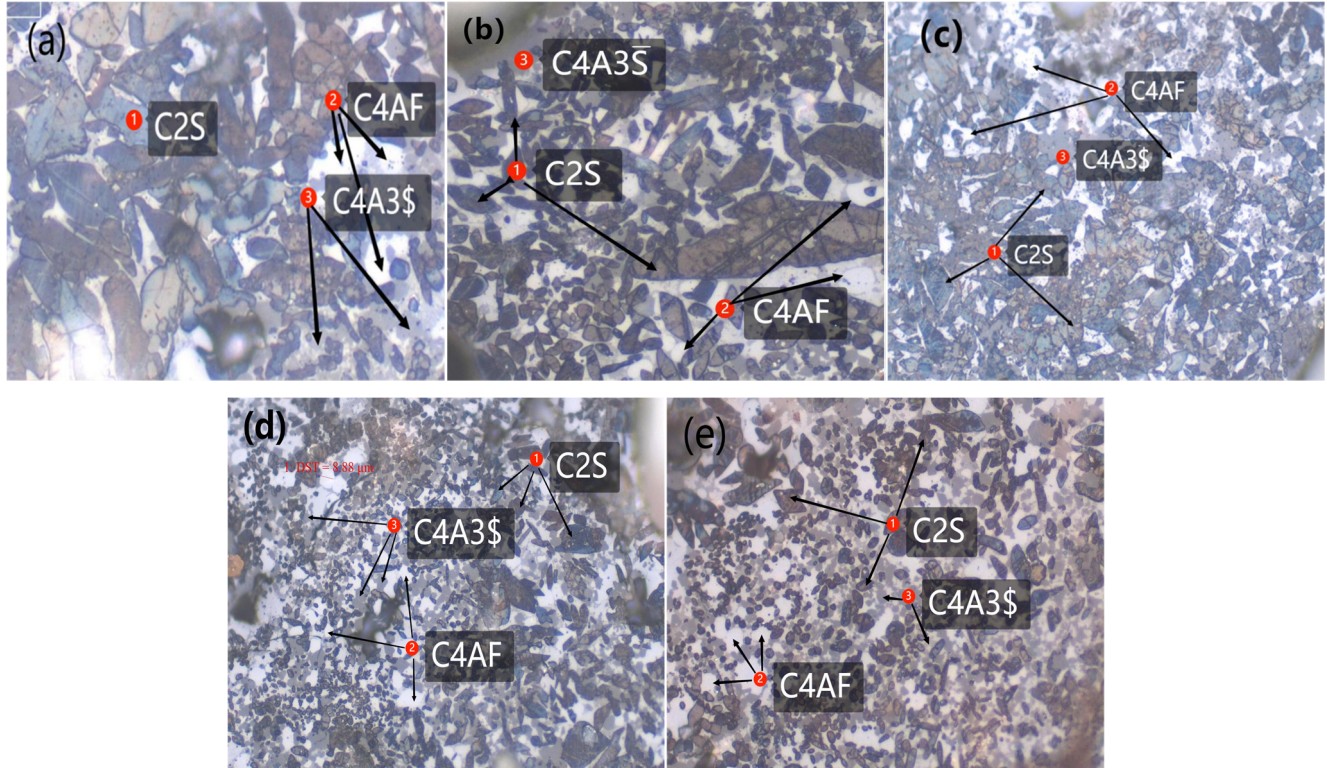

**Figure 3.** Petrographic images of HBSC clinker (**a**) GF-1 × 400, (**b**) GF-2 × 400, (**c**) GF-3 × 400, (**d**) GF-4 × 400, and (**e**) GF-5 × 400.

### 3.2.3. Mineral Composition

The mineral composition of HBSC clinker containing brownmillerite sintering at 1330 °C for 30 min was analyzed by qualitative XRD (QXRD). QXRD patterns are shown in Figure 4. The corresponding mineral compositions calculated through QXRD are listed in Table 9. The main minerals in HBSC clinker containing brownmillerite were $C_4A_3\$$, $C_2S$, brownmillerite, gehlenite ($C_2AS$), perovskite and $C_3A$. As shown in Table 9, $C_4A_3\$$ and brownmillerite were slightly higher and much lower, respectively, than the theoretical mineral compositions shown in Table 2. It has been reported that either $Fe^{3+}$ or $Al^{3+}$ could occupy $al^{3+}$ position in $C_4A_3\$$ structures to form a $C_4A_{3-x}F_x\$$ solid solution. The replacement amount of $Fe^{3+}$ could range from 2% to 10% or even higher [15–18,26–29]. Substitution of $Fe^{3+}$ for $Al^{3+}$ in $C_4A_3\$$ led to a lower content of actually generated brownmillerite than the theoretical calculation value. Meanwhile, some $Al^{3+}$ existed in the form of $C_3A$ (Table 9). Overall, the $C_4A_3\$$ increased to more than the theoretical mineral composition with the increase in the designed brownmillerite content in HBSC clinker; that is, brownmillerite promoted the formation of $C_4A_3\$$, and also had a specific stabilizing effect for $C_4A_3\$$. The content of $C_2S$ differed little from the theoretical content, as shown in Tables 2 and 9. However, $\beta$-$C_2S$ content tended to decrease with the increase in designed brownmillerite content, while $\gamma$-$C_2S$ content showed an increasing trend. $Fe_2O_3$ can interact with $\beta$-$C_2S$ to form an unstable solid solution, further reduced to FeO in a reductive atmosphere to form $CaO \cdot FeO \cdot SiO_2$ (CFS), thus destroying $\beta$-$C_2S$ lattices [30,31]. As is well

known, $\gamma$-C$_2$S has poor hydration activity, and the increase in $\gamma$-C$_2$S with the increase in designed brownmillerite content indicated that brownmillerite destroyed the $\beta$-C$_2$S lattice and might affect the hydration of HBSC.

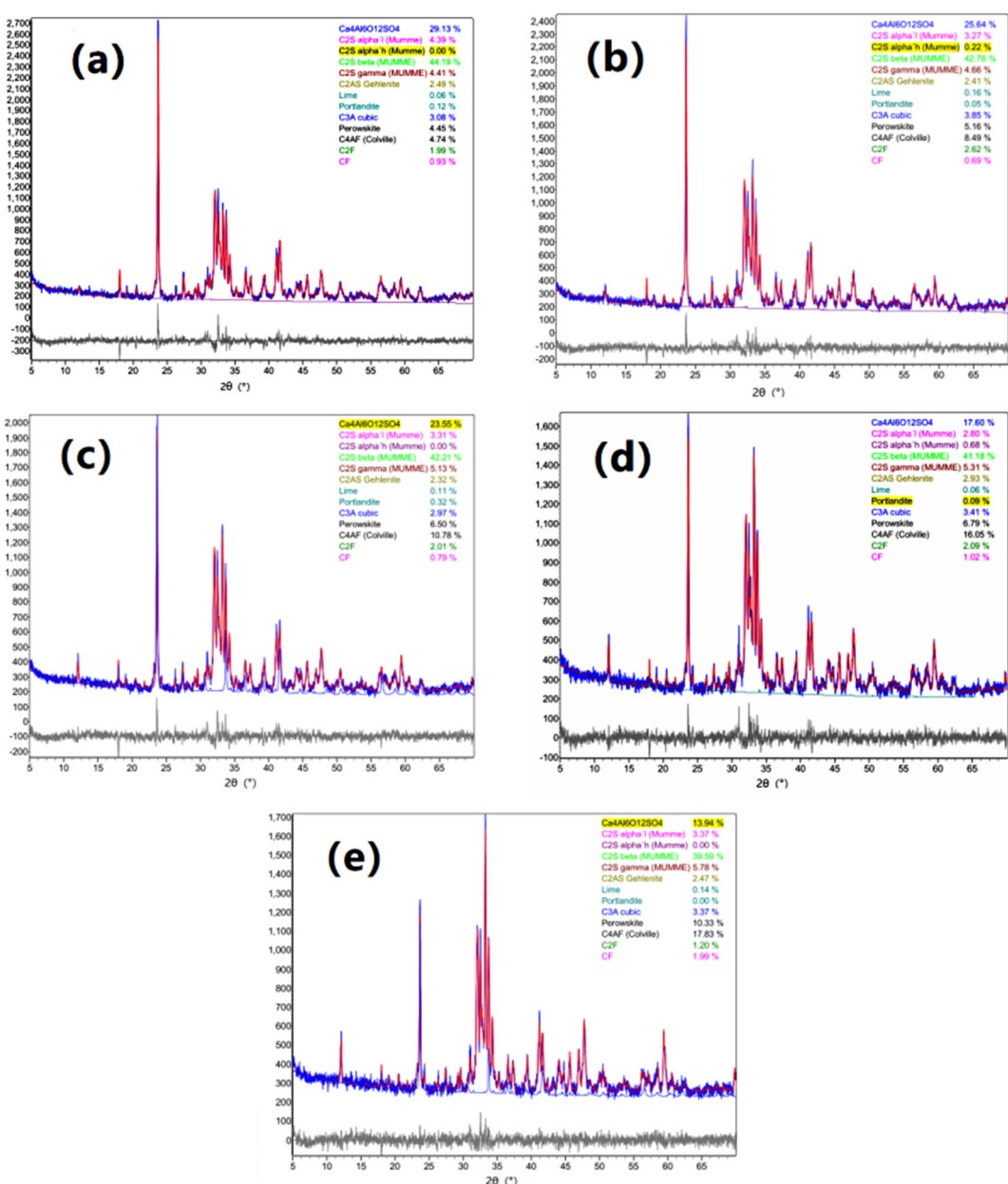

**Figure 4.** QXRD patterns of HBSC clinker containing brownmillerite (**a**) GF-1, Rwp = 5.80%, (**b**) GF-2, Rwp = 5.88%, (**c**) GF-3, Rwp = 5.90%, (**d**) GF-4, Rwp = 6.07%, and (**e**) GF-5, Rwp = 6.22%.

**Table 9.** Main mineral composition of HBSC clinker through QXRD patterns (wt%).

| No. | C$_4$A$_3$$ | C$_2$S | | | | C$_2$AS | C$_3$A | Brownmillerite | | | |
| --- | --- | --- | --- | --- | --- | --- | --- | --- | --- | --- | --- |
| | | $\alpha$ | $\beta$ | $\gamma$ | $\Sigma$ | | | C$_4$AF | C$_2$F | CF | $\Sigma$ |
| GF-1 | 29.13 | 4.39 | 44.19 | 4.41 | 52.99 | 2.49 | 3.08 | 4.74 | 1.99 | 0.93 | 7.66 |
| GF-2 | 25.64 | 3.49 | 42.78 | 4.66 | 50.93 | 2.41 | 3.85 | 8.49 | 2.62 | 0.69 | 11.80 |
| GF-3 | 23.55 | 3.31 | 42.21 | 5.13 | 50.65 | 2.32 | 2.97 | 10.78 | 2.01 | 0.79 | 13.58 |
| GF-4 | 17.60 | 3.48 | 41.18 | 5.31 | 49.97 | 2.93 | 3.41 | 16.05 | 2.09 | 1.02 | 19.16 |
| GF-5 | 13.94 | 3.37 | 39.59 | 5.78 | 48.74 | 2.47 | 3.37 | 17.83 | 1.20 | 1.99 | 21.02 |

### 3.3. Mechanical Performance of HBSC Containing Brownmillerite

Figure 5 shows the compressive strength of HBSC pastes after curing for different time periods. After 1 day, 3 days and 7 days, the compressive strengths decreased with the increase in brownmillerite content, which was attributed to the decrease in the total amount of early-strength clinker minerals ($C_4A_3\$$ and $C_4AF$, in Table 8) [32,33]. After longer periods (21 days and 28 days), the compressive strength of HBSC pastes increased when designed brownmillerite in HBSC clinker was below 25%, but decreased when designed brownmillerite in HBSC clinker was above 25%. Relatively higher content of β-$C_2S$ promotes the strength development of HBSC containing brownmillerite. However, β-$C_2S$ would be destroyed and γ-$C_2S$ would be increased with the increase in designed brownmillerite, which significantly decreased the later compressive strength of HBSC pastes. A proper amount of $C_4AF$ can ensure the early strength and the strength development of HBSC.

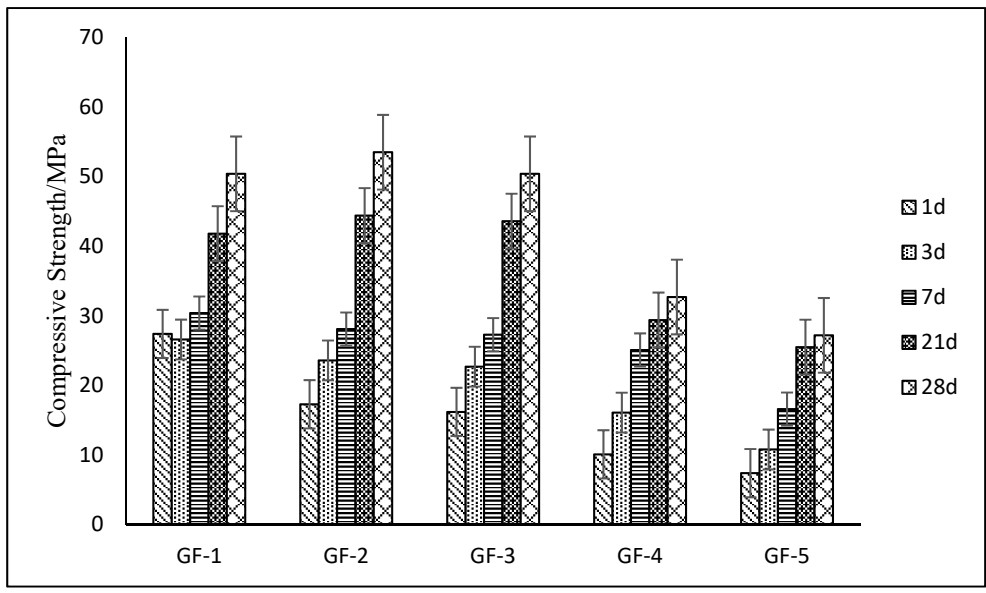

**Figure 5.** Compressive strength of HBSC pastes.

### 4. Conclusions

The effect of brownmillerite phase content on the sintering of HBSC clinker prepared with industrial raw materials is complex. The high brownmillerite phase promoted the formation of clinker minerals and stabilized the $C_4A_3\$$ through the substitution of $Fe^{3+}$ for $Al^{3+}$ in $C_4A_3\$$, which increased the actual $C_4A_3\$$ content and decreased the $C_4AF$ content compared to that of designed theoretical mineral composition. At the same time, it changed the crystal form and morphology of $C_2S$ minerals, and its existing forms and morphologies also changed significantly. The early compressive strength of HBSC pastes decreased with the increase in brownmillerite content due to the decrease in the total amount of early-strength clinker minerals. A proper amount of brownmillerite in the HBSC clinker ensured the early strength and the strength development of HBSC pastes.

**Author Contributions:** Conceptualization, X.C., J.L. and Z.L. (Zhongyuan Lu); methodology, X.C., J.L. and Z.L. (Zhongyuan Lu); software, X.C. and J.J.; validation, X.C. and J.L.; formal analysis, Z.L. (Zhongyuan Lu) and Y.X.; investigation, X.C.; resources, X.C.; data curation, X.C.; writing—original draft preparation, X.C.; writing—review and editing, X.C., J.L., S.N. and Z.L. (Zhenyu Lai); visualization, X.C., Y.N. and H.L.; supervision, Z.L. (Zhongyuan Lu) and Y.X.; project administration, X.C. and J.L.; funding acquisition, Z.L. (Zhongyuan Lu) and Y.X. All authors have read and agreed to the published version of the manuscript.

**Funding:** This work was supported by Sichuan Science and Technology Program (No. 2019ZDZX0024), the National Natural Science Foundation of China (No. 51402246), Natural Science Foundation by Southwest University of Science and Technology (No. 19zx7126).

**Institutional Review Board Statement:** The study did not require ethical approval.

**Informed Consent Statement:** The study did not involve humans.

**Data Availability Statement:** Not applicable.

**Conflicts of Interest:** The authors declare no conflict of interest. The funders had no role in the design of the study; in the collection, analyses, or interpretation of data; in the writing of the manuscript, or in the decision to publish the results.

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
