# Peer review of "The Role of Brownmillerite in Preparation of High-Belite Sulfoaluminate Cement Clinker"

_applsci, doi:10.3390/app12104980_

Round 1

Reviewer 1 Report

Please see my comments in attached file. Good paper, I recommend only minor revisions. Best regards!

Author Response

Dear reviewer,

Thank you very much for your careful review and constructive suggestions about our manuscript. Those comments are helpful for us to revise and improve our paper. We have studied the comments carefully and tried our best to revise and improve the manuscript according to the comments. The main corrections in the paper and the responds to your comments are listed as flowing.Please see the attachment.

Best regards,

Xuemei Chen

Reviewer 2 Report

For this study, we prepared high-belite-sulfoaluminate cement (HBSC) clinker containing brownmillerite. Thermal analysis, petrographic analysis and quantitative X-ray diffraction (QXRD) were used to determine the effect of brownmillerite on the mechanical performance of clinker sintering, clinker mineral and HBSC.

In this study, there are no special problems with the experimental design or material selection. However, the originality of this study is questionable. This is because this thesis is only basic data that can be used in the manufacture of cement clinker.

In order to raise the scientific level of this study, more experimental volume and deeper scientific inquiry are required. I do not think this paper will be of great interest to readers. This is because most of the predictions of the experimental results are possible.

The introductory part of this thesis does not know what the originality of this thesis is. And the development of the logic is not natural.

In terms of experimental design and content analysis, the overall content is simple and does not seem to contribute significantly to the cement field. Also, the conclusion part is really weakly written.

Therefore, I do not want this paper to be published in this journal.

Author Response

(The authors gave the same response as above.)

Reviewer 3 Report

Authors have made a complete microstructural characterization of high belite-sulphoaluminate cement clinkers. The conclusions specified agree with the obtained results. The reviewer wants to congratulate the authors for the work done.

Author Response

Dear  reviewer,

Thank you very much for your careful review about our manuscript and your affirmation of our work .

Best regards,

Xuemei Chen

Round 2

Reviewer 2 Report

I changed my mind after seeing the answers from the authors. My misunderstanding was resolved through the authors' answers. And through the efforts of the authors, the quality of the paper has improved a lot.